# Exploiting Mitochondrial Vulnerabilities to Trigger Apoptosis Selectively in Cancer Cells

**DOI:** 10.3390/cancers11070916

**Published:** 2019-06-29

**Authors:** Christopher Nguyen, Siyaram Pandey

**Affiliations:** Department of Chemistry and Biochemistry, University of Windsor, Windsor, ON N9E 3P4, Canada

**Keywords:** mitochondria, apoptosis, oxidative phosphorylation, reactive oxygen species, metabolic reprogramming, electron transport chain, sensitization, chemoresistance

## Abstract

The transformation of normal cells to the cancerous stage involves multiple genetic changes or mutations leading to hyperproliferation, resistance to apoptosis, and evasion of the host immune system. However, to accomplish hyperproliferation, cancer cells undergo profound metabolic reprogramming including oxidative glycolysis and acidification of the cytoplasm, leading to hyperpolarization of the mitochondrial membrane. The majority of drug development research in the past has focused on targeting DNA replication, repair, and tubulin polymerization to induce apoptosis in cancer cells. Unfortunately, these are not cancer-selective targets. Recently, researchers have started focusing on metabolic, mitochondrial, and oxidative stress vulnerabilities of cancer cells that can be exploited as selective targets for inducing cancer cell death. Indeed, the hyperpolarization of mitochondrial membranes in cancer cells can lead to selective importing of mitocans that can induce apoptotic effects. Herein, we will discuss recent mitochondrial-selective anticancer compounds (mitocans) that have shown selective toxicity against cancer cells. Increased oxidative stress has also been shown to be very effective in selectively inducing cell death in cancer cells. This oxidative stress could lead to mitochondrial dysfunction, which in turn will produce more reactive oxygen species (ROS). This creates a vicious cycle of mitochondrial dysfunction and ROS production, irreversibly leading to cell suicide. We will also explore the possibility of combining these compounds to sensitize cancer cells to the conventional anticancer agents. Mitocans in combination with selective oxidative-stress producing agents could be very effective anticancer treatments with minimal effect on healthy cells.

## 1. Background of Mitochondria and Cancer

Despite many advances in cancer treatment approaches, cancer remains the leading cause of death in both Canada and the United States [1,2]. In Canada, of 206,200 projected cancer incidences, 1 in 4 diagnosed patients are expected to succumb to the disease in 2019 [1]. In the United States, of 1,762,450 projected diagnoses in 2019, about 1 in 3 diagnosed patients are expected to succumb to the disease [2]. Therefore, it is critical to discover new cancer-selective treatments that are both more effective and less toxic to patients.

Cancer cells are notorious for their enhanced ability to proliferate in suboptimal conditions through the accumulation of many genetic mutations. They are able to manipulate metabolic and immunogenic pathways to adapt and operate under difficult conditions [3]. Indeed, metabolic reprogramming in cancer cells has been identified as a hallmark of cancer and a potential vulnerability that can be targeted to fight this disease [4].

Apoptosis, the complex and physiological process of cell death, removes both unwanted cells and those with damaged DNA. In doing so, this process can act as a safeguard against the development of abnormal cells. Apoptosis can be triggered by DNA damage, oxidative stress, growth factor deprivation, and mitochondrial depolarization [5]. Knowledge of the biochemical mechanisms of apoptosis has led to the development of therapies targeting cancer cells to induce cell death [5]. Various processes, including those mentioned previously, can trigger pathways by which apoptosis can initiate and progress. In the intrinsic pathway, internal signals such as DNA damage can lead to a cascade of events, resulting in activation of the Bcl-2 family of proteins, which in turn activate Bax-like proteins to permeabilize the membrane and release cytochrome c [5]. Cytochrome c release will lead to the activation of caspase proteins to induce apoptosis. In the extrinsic pathway, external signals such as tumour necrosis factor (TNF) can lead to caspase activation and apoptosis [5]. The exploitation of vulnerabilities in cancerous cells, including oxidative stress susceptibility and mitochondrial membrane destabilization, could be used to develop novel therapeutic agents that trigger apoptosis and eradicate the disease [6,7]. Researchers have aimed to exploit cancer cell vulnerabilities using a variety of approaches including identification of tumour-suppressor genes, discovery of novel compounds, and development of technologies such as nanoparticles to selectively identify targets that can be effective and selective for cancer [8,9,10]. Unfortunately, cancer cells develop resistance to many chemotherapies by resisting apoptosis and/or exporting the drugs outside the cell. This is achieved by disrupting pro-apoptotic proteins, reducing the function of caspases, and impairing death receptor signaling [5]. 

Cancer therapeutics including DNA damaging agents (e.g., cisplatin, doxorubicin, or 5-fluorouracil) and tubulin modifying agents (e.g., paclitaxel) have been developed to induce apoptosis in cancer cells. However, these drugs have limitations due to their non-selective nature and extreme toxicity to healthy tissues. Further, cancer cells develop resistance to these drugs and patients may experience severe side effects with no significant impact on the cancer [11,12,13]. In addition, these genotoxic treatments can lead to DNA damage in normal tissues, increasing the risk of further cancer development. Thus, we are essentially using cancer-inducing compounds in our attempts to treat cancer.

In order to proliferate in hostile conditions, cancer cells undergo metabolic reprogramming to meet their energetic needs. In the 1920s, Otto Warburg observed what he termed the ‘Warburg Effect’, which described the production of excessive lactate by cancer cells in the presence of oxygen as a result of ‘oxidative glycolysis’ [14,15]. However, this has led to the misunderstanding that cancer cells have mitochondrial damage and metabolic inefficiencies, forcing them to rely on excessive glycolysis for their energy production [16]. Contrary to this, a large breadth of work has shown that mitochondria and mitochondrial respiration remains undamaged in many cancers [17]. Interestingly, in ovarian cancer, the microenvironment of cancer cells directly impacts their metabolic reprogramming [18]. Peripheral cells of a tumour spheroid exposed to normal oxygen levels relied heavily on aerobic glycolysis and were proliferative cells. In contrast, internal cells were poorly vascularized with poor access to oxygen and glucose, leading to cell quiescence and heavy dependence on mitochondrial oxidative phosphorylation (OxPhos) for the majority of their ATP production [19,20]. This study highlights that the fact that while standard chemotherapy was effective at killing proliferative cells, the quiescent cell population is resistant and causes tumour regeneration. In order to properly combat cancer, a multi-faceted approach targeting mitochondrial vulnerabilities should be considered to ensure complete tumour elimination. In fact, recent observations have indicated that mitochondria actually support and play a critical role in tumourigenesis through their metabolic reprogramming, oxidative signaling, reactive oxygen species generation, and production of oncometabolites [21,22,23,24,25].

Cancer cells have higher metabolic needs and antioxidant defenses compared with healthy cells. They rely heavily on aerobic glycolysis to meet their energy needs and, as a result, upregulate glucose transporters to meet their demands [26,27]. Furthermore, aerobic glycolysis leads to the production of large amounts of lactate and pyruvate, causing increased acidity in the cytoplasm in cancer cells. Thus, their mitochondria are hyperpolarized compared with those of normal cells [27]. This hyperpolarization can also be the result of increased intracellular Ca^2+^ levels and upregulation of anti-apoptotic Bcl2 protein [28,29] and/or increased apoptosis evasion [30].

While the mitochondria are the organelle responsible for ATP generation, they contain a number of pro-apoptotic factors such as cytochrome c, endonuclease G, and apoptosis inducing factor (AIF), which can induce a cell suicide program if they are released outside. Release of cytochrome c in the cytosol leads to its association with apoptosis protease activating factor 1 (APAF-1) and Caspase-9, eventually forming the apoptosome that activates Caspase-3 and executes apoptosis [31]. In contrast, AIF-initiated apoptosis is caspase-independent, and works through chromatin condensation and DNA fragmentation [32]. The presence of such pro-apoptotic proteins in the mitochondria puts a spotlight on the organelle as an interesting target for cancer therapy research.

Mitochondria play a central role in the apoptosis induction process. It would be appropriate to say that each mitochondrion acts as a “self-destruct” button for the cell. Differences between cancerous and non-cancerous mitochondria can be targeted to allow for the release of pro-apoptotic factors to induce apoptosis selectively in cancer cells. In this review, we will discuss some of the recent advances in the development of therapeutic modulators targeting mitochondrial vulnerabilities.

## 2. Targeting Mitochondrial Vulnerabilities

Cancer cells may rely heavily on OxPhos in addition to aerobic glycolysis for their excessive energy needs [33,34]. Further, mitochondrial membranes of cancer cells are hyperpolarized relative to normal healthy cells and are poorly assembled [35]. Mitochondrial-targeting treatments thus have the potential to specifically target cancer while selectively sparing normal healthy cells. Mitochondria also play a fundamental role in reactive oxygen species and oxidative stress defense, energy production, and the induction of apoptosis. These mechanisms and functions may be altered in cancer and serve as specific markers to be targeted by therapeutics.

By exploiting cancer cell differences in needs and functions, healthy cells can be spared, leading to treatments that avoid adverse side-effects, as observed in many chemotherapeutics today. For example, Weinberg and Chandel postulate that while cancer cells may simply turn to aerobic glycolysis, there are other reasons supporting targeting the mitochondria [19]. The first being that poorly perfused tumours may have limited access to aerobic conditions and glucose. This forces cancer cells to rely on mitochondrial ATP generation, which can be specifically targeted by therapies [36,37]. The second being that some cancer cells indeed show a heavy dependence on OxPhos for their ATP needs [37,38]. In both scenarios, mitochondrial-targeting treatments can disrupt OxPhos machinery and lead to cancer cell death. Finally, drugs targeting and inhibiting mitochondrial ATP production may sensitize cancer cells to aerobic glycolysis-targeting drugs and enhance their action. Further, when the mitochondria are targeted, leakage of pro-apoptotic factors from mitochondria will regardless result in activation of apoptotic. Further, when the mitochondria are targeted, leakage of pro-apoptotic factors will result in activation of apoptosis, supporting the notion of mitochondrial-targeting drug synergy.

### 2.1. Induction of Oxidative Stress as a Target for Cancer Treatment

Reactive oxygen species (ROS) are radicals containing a single unpaired electron in their outermost electron shell [39]. At lower levels, ROS can be advantageous in promoting proliferation and signaling [40,41]. However, at higher levels, ROS can induce oxidative stress leading to cell death [40]. Cancer cells, unlike healthy cells, require a higher concentration of ROS to supplement increased proliferation rates. It has been hypothesized that cancer cells can take advantage of the resulting augmented DNA-damage to promote further mutation and tumourigenesis [41,42]. Further, the induction of oxidative stress via chemotherapy or radiation therapy on cancer cells may result in DNA damage-induced cell death [43,44,45,46]. Because of their metabolic changes, a shift to increased ROS in the microenvironment allows for progressive mutation towards a metastatic state [40]. As a result, cancer cells have upregulated antioxidative defenses to survive in their enhanced oxidizing environment.

The production of ROS in the mitochondria is inevitable because of its generation as a by-product in OxPhos. ROS presence plays a role in alteration of mitochondrial dynamics [47] and, as a result, is efficiently quenched in normal cells using an antioxidative defense system that includes superoxide dismutase and glutathione peroxidase [48,49]. If not quenched or eliminated, excessive oxidative stress can cause further dysfunction of mitochondrial proteins, leading to augmented production of ROS, creating a vicious cycle of mitochondrial damage and oxidative stress. This will eventually lead to the collapse of the mitochondrial membrane potential, permeabilization of the membrane, and induction of apoptosis [7].

### 2.2. Mitochondrial Metabolic Reprogramming as a Target for Cancer Treatment

Cancer cells rely heavily on aerobic glycolysis for the bulk of their energy needs and turn to OxPhos only in situations where oxygen or glucose access is limited. The use of glycolytic targeting therapies may help to reduce the proliferation of cancer cells. Upregulation of aerobic glycolysis is a result of increased expression of oncogenes (such as MYC and KRAS) and deregulation of the P13K signaling pathway [50,51,52]. Owing to excessive glycolysis and lactic acid production, mitochondria are hyperpolarized [26]. As a result of metabolic reprogramming by cancer cells, specific aspects of cancer cell metabolism may be distinct from normal healthy cells. These differences can be targeted by therapies to enhance selectivity for cancer cells and to avoid adverse side effects due to targeting of healthy cells. Specific targets will be discussed in the later sections below.

Recent work on targeting of mitochondrial reprogramming vulnerabilities has focused on many different approaches, including inhibition of upregulated metabolic proteins found only in cancer cells [53], targeting mitochondrial oxidative phosphorylation or respiration [54,55], or through the induction of antibiotics-induced mitochondrial dysfunction [56].

### 2.3. Sensitization and Reversal of Chemoresistance by Targeting the Mitochondria

Cancer cell mitochondria contain higher amounts of anti-apoptotic Bcl2 family of proteins to evade apoptosis and as such are resistant to anti-cancer drugs. The depletion of mitochondrial DNA (mtDNA) has been reported in numerous cancers in vivo and has been implicated in increasing the expression of anti-apoptotic genes, such as Bcl2. This activates pro-survival enzymes, leading to resistance to chemotherapy-mediated apoptosis [57,58,59,60,61]. Owing to a dependence on aerobic glycolysis, cancer cells are able to resist apoptosis through upregulation of regulatory enzymes [62]. For example, hexokinase II has been shown to increase lactate production, cell proliferation, resistance to drugs, and invasion [63]. Lactate production allows for cancer cells to maintain a slightly acidic micro-environment and enhance their survival through pathways such as using lactate as an antioxidant [64,65,66].

There have been many reported cases of chemosensitization by targeting the mitochondria of cancer cells. One example is the drug metformin, which targets complex I of the electron transport chain in cancer cell mitochondria. Metformin is a novel and exciting development showing promise as an anti-cancer agent and adjuvant through its ability to sensitize cancer cells, particularly in ovarian cancer [67,68,69,70]. As mentioned previously, internal quiescent ovarian cancer cells develop a resistance to chemotherapeutic treatments that are effective against proliferative cells, requiring the addition of OxPhos inhibitor therapy to prevent cancer relapse [18]. In addition, multi-drug resistance to doxorubicin was overcome by utilizing doxorubicin-loaded micelles to allow for the chemotherapy to reach the mitochondria more effectively [71]. 

## 3. Strategies to Target Mitochondria

Indeed, a large breadth of work has been put into the identification of mitochondrial vulnerabilities and the identification of mitochondrial differences in cancerous and non-cancerous cells. In doing so, specific targeting approaches of cancer cell mitochondria have been developed. In this section, we will summarize the advances made recently into the development of mitocans as potential therapeutics for cancer. Olivas-Aguirre et al. recently published a great review on mitochondrial-targeted therapies for T cell acute lymphoblastic leukemia focusing on specific targets for mitocans [72]. We will highlight several mitochondrial targets and expand the breadth to include all cancer types. We have tabulated all treatments reviewed in Table 1 and visualized their targets in Figure 1.

### 3.1. Targeting Mitochondrial Interacting Hexokinase II

Cancer cells rely heavily on aerobic glycolysis and, as a result, over-express hexokinases (HK) that catalyze the first step of glucose metabolism [131,132]. Of the four HK isoforms, hexokinase II (HKII) plays a critical role in cancer cell survival and proliferation. At the outer mitochondrial membrane, HKII binds to voltage-dependent anion channel 1 (VDAC1) and facilitates its interaction with adenine nucleotide translocase on the inner mitochondrial membrane [133,134]. This interaction is able to couple aerobic glycolysis with OxPhos and create a working relationship between the two metabolic processes, allowing HKII to exchange ADP for ATP from the mitochondria and increase the rate of glycolysis [72,134]. Targeting HKII could lead to uncoupling and stunting of aerobic glycolysis, leading to cancer cell death.

Recent research has focused on agents that are capable of inhibiting HKII and preventing the progression of glucose metabolism. Some commonly used inhibitors include glucose analogs 2-deoxy-D-glucose (2-DG) and 3-bromopyruvate (3-BPA), which have been shown to induce cytotoxic effects on acute lymphoblastic leukemia while enhancing the efficacy of prednisone treatment [73,74]. An epidermal growth factor receptor-targeted liposomal formulation of 3-BPA showed improved permeability, HKII inhibition, and cytotoxicity compared with conventional aqueous 3-BPA formulations [75]. However, a recent study indicated that 3-BPA was still effective in killing HKII knockout cells, thus suggesting that in addition to HKII, the efficacy of 3-BPA may be also dependent on tumour microenvironment and glucose availability [76]. A cytotoxic peptidoglycan complex and a synthetic peptide derived from *Lactobacillus casei* were both able to impair the entire metabolism of tumour cells via displacement of HKII from the mitochondrial membrane [77]. Importantly, these compounds were cytotoxic to cancer cells while stimulating glycolysis in healthy noncancerous cells, presumably because of the cancer cell dependence on HKII binding to the mitochondrial membrane. Benserazide (Benz), originally designed to treat Parkinson’s Disease, is another inhibitor of HKII that was able to selectively reduce glucose uptake, lactate production, and intracellular ATP levels, leading to the dissipation of the mitochondrial membrane potential (MMP) and apoptosis [78].

There have also been efforts to genetically target HKII to kill cancer cells. Signal transducer and activator of transcription 3 (STAT3) is an oncogene playing critical roles in tumour development, angiogenesis, and metastasis [135]. STAT3 is known as a downstream factor from rapamycin (mTOR) and a regulator of HKII; thus, the mTOR–STAT3–HKII pathway is an interesting target for glycolysis inhibition in cancer cells [79]. Rapamycin treatment and siRNA downregulation of STAT3 were shown to directly decrease glucose consumption and downregulate HKII [79]. Knockdown of HKII was also achieved using microRNA-143 (miR-143) overexpression, and led to the promotion of cancer cell apoptosis through inhibition of glucose metabolism and proliferation [80]. In the same regard, overexpression of miR-218 downregulated HKII expression, facilitated by the cell-promoting oncogene Bmi1, in a novel miR-218/Bmi1/HKII axis [81,82].

Interestingly, HKII has also been implicated as a target to sensitize cancer cells to chemotherapy and radiotherapy. Upregulation of HKII has been shown to modulate resistance to rituximab, and inhibition of HKII (via 2-DG, discussed above) resulted in sensitization of these cells to rituximab, as indicated by decreased MMP and cell viability [136]. HKII depletion has also pushed cancer cells to resort to OxPhos, sensitizing cancer cells to be targeted by complex I inhibitor metformin to facilitate apoptosis [112]. Other studies have shown that targeting both HKII and its regulators led to radiation treatment sensitization [137,138].

Of these HKII targeting compounds, Benz (clinical trial number: NCT02741947) and metformin usage in human cancers (clinical trial number: NCT03477162) has progressed to clinical studies. However, it is important to note that Benz completed phase 4 clinical trials testing for usage in Parkinson’s Disease and not cancer.

### 3.2. Targeting Voltage-Dependent Anion Channel 1

Voltage-dependent anion channel 1 (VDAC1) is situated on the outer mitochondrial membrane and allows for the transfer of many compounds including metabolites, fatty acids, Ca^2+^, ROS, and cholesterol across the mitochondrial membrane [139]. 

By targeting VDAC1, it is possible to hinder cancer cell apoptosis evading mechanisms. VDAC1 plays a central role in mitochondrial apoptosis, interacting with anti-apoptotic proteins such as Bcl2 or Bcl-xL, as well as apoptosis-suppressing HKII. Synthetic peptides based on VDAC1 binding sites were used to inhibit Bcl2, Bcl-xL, and HKII, preventing their association with VDAC1 and thus inhibiting cancer cells’ ability to evade apoptosis [83]. VDAC1-based peptides, Antp-LP4 and N-Ter-Antp, were engineered and were highly efficacious on peripheral blood mononuclear cells (PBMCs) from leukemia patients, but spared healthy patient PBMCs [84]. These peptides were also highly effective in inducing cell death in a variety of cancer cell lines [85]. The same research team further designed peptides with increased specificity, including the R-Tf-D-LP4 peptide, which contain a transferrin receptor internalization sequence (Tf) to promote the targeting of overexpressed transferrin receptor in cancer cells [86]. This peptide was widely successful, however, common metabolic enzymes such as GLUT-1, GAPDH, and citrate synthase were targeted, potentially leading to adverse side effects in healthy cells and complications in clinical trials. 

Taking a genetic approach to targeting this protein, siRNA silencing of VDAC1 resulted in decreased MMP and ATP levels in cancer cell lines, as well as a drastic reduction of tumour burden on lung cancer xenografted mice [87]. An antifungal drug, itraconazole, was shown to target VDAC1 and inhibit mTOR activity and cell proliferation through modulation of mitochondrial metabolism, leading to apoptosis [88]. However, VDAC1 knockdown may have toxic effects owing to their expression in healthy cells, and more studies on toxicity should be done [139].

In addition, the VDAC1–HKII complex can be targeted to trigger apoptosis in cancer cells overexpressing HKII. In a recent study, fenofibrate interrupted the binding of HKII to VDAC1 and reprogrammed the metabolic pathway in oral squamous cell carcinoma [89]. Many other compounds that are capable of destroying the VDAC–HKII bond have been highlighted by Magri et al. [140], including clotrimazole [90], 3-BR [141], 2DG [142], oroxillin A, lonidamine, arsenites, and steroid analogs [91].

Fenofibrate (clinical trial number: NCT01965834) and lonidamine (clinical trial number: NCT00435448) entered clinical trials, but were terminated at phase II and III, respectively.

### 3.3. Targeting Bcl2

The family of anti-apoptotic B-cell lymphoma 2 (Bcl-2) proteins present an interesting target for cancer therapies because of their role in promoting cell survival inhibition of apoptosis [143]. Initial attempts to develop agents to target Bcl-2 aimed to decrease Bcl-2 levels through the delivery of RNA antisense molecules [144]. This study highlighted molecules that have significant anticancer effects and have entered clinical trials, including Oblimersen (G3139/Genasense) [92], PNT2258 (NCT0222696) [93], and SPC2996 (NCT00285103) [94]. BH-3 mimetics imitate protein–protein interactions between BH-3 domains and Bcl-2 family members, and have been used to displace bound Bcl-2 protein from pro-apoptotic partners, leading to cancer cell death [144]. ABT-737, the first BH-3 mimetic, efficiently induced apoptosis in lymphoma and leukemia cell lines [95,96], leading to the development of orally bioavailable ABT-263 (navitoclax) and ABT-199 (venetoclax), selective to Bcl2, which have shown promise in clinical studies [97,98]. A novel anthraquinone BH-3 mimetic named compound 6 was identified to be the first small molecule to induce apoptosis in melanoma cancer cells by binding to Bcl-2, Mcl-2, and phosphorylated Mcl-2 [99].

Mitochondrial pyruvate kinase M2 isoform (PKM2) can translocate to the mitochondria and interact with Bcl-2 to regulate oxidative stress-induced apoptosis via stabilization of Bcl-2 [100]. siRNA knockdown of PKM2 resulted in decreased viability and increased apoptosis in hepatocellular carcinoma, ovarian carcinoma, and colon cancer cells [101]. On the other hand, somatostatin structural analogue TT-232 can interact with PKM2, translocate it to the nucleus, and trigger caspase-independent cell death [102]. 

Owing to their role as anti-apoptotic proteins, the Bcl-2 family of proteins has attracted great attention for their potential to be targeted to reverse chemoresistance. With platinum-based drugs such as cisplatin, the usage of Bcl-2 inhibitors such as ABT-737 led to the reversal of platinum resistance in ovarian cancer, sensitizing these cells to cisplatin and leading to apoptosis [145]. MicroRNAs (miRNAs) have also been used to modulate multidrug resistance in cancer cells to sensitize them to chemotherapies, such as ectopically expressed miR-181b [103] and mir-630 [104], showing the ability to overcome chemoresistance and leading to a favourable response to cisplatin treatment. Bcl-2 inhibition by ABT-199 treatment showed synergistic viability decreases when used in combination with doxorubicin in breast cancer, supporting the notion that Bcl-2 inhibition can sensitize cancer cells to chemotherapy [146].

ABT-199 (clinical trial number: NCT03181126), ABT-263 (clinical trial number: NCT03504644), and TT-232 (clinical trial number: NCT00422786) are all currently undergoing phase I or II clinical trials.

### 3.4. Targeting Electron Transport Chain Complexes

An interesting approach to treatment involves engineering mitocans (selective, mitochondrial-targeting anticancer agents) that are able to target electron transport chain (ETC) complex proteins [147]. Treatments designed to target cancer cell OxPhos may shut down important machinery in cancer cell metabolism, sensitizing them to aerobic glycolysis targeting agents or potentially facilitating the leakage of mitochondrial pro-apoptotic proteins.

Many mitocans targeting complexes I, II, and III have shown great efficacy and selectivity for cancer. Sorafenib (nexavar) inhibits the activity of complex II and complex III of the ETC along with ATP synthase, leading to the stabilization of serine-threonine protein kinase PINK1, which in turn induces Parkin-mediated apoptosis [105]. A derivative of tamoxifen, MitoTam, was conjugated with membrane-permeable cation triphenylphosphonium (TPP), allowing for the localization of MitoTam in the mitochondria to target complex I [106]. MitoTam suppressed Her2^high^ tumours and avoided toxicity to healthy cells. Similarly, other mitocans like TPP-peptide [107], artemisinin-TPP [108], and green titania (G-TiO_2-x_) conjugated to TPP [109] have been shown to selectively kill cancer cells. Reservatrol has been shown to act on and inhibit complexes I and III and act as a pro-oxidant, leading to cell death in cancer [110,111]. 

Metformin has gained a great deal of interest for its anti-cancer efficacy. Metformin directly inhibits complex I of the mitochondria ETC [113]. Originally a drug used to treat diabetes, metformin has burst onto the scene as a new and exciting mitocan capable of targeting cancer very selectively [148]. Of great interest is the ability of metformin to act as an adjuvant and sensitize cancer cells to other treatments, including therapies such as ABT-737 [67,68,69,70,149,150,151]. As such, metformin provides solid evidence that targeting ETC complexes allows for an efficacious mono or adjuvant treatment for cancer.

We have previously demonstrated that pancrasitatin (PST) analogs induced apoptosis at extremely low EC_50_ levels in a wide variety of cancers with a much higher efficacy than standard chemotherapeutics [115]. Importantly, our mitocans were highly selective for cancer, with minimal toxicity on normal healthy cells in vitro (both 2D and 3D spheroid culture models) and in vivo. In order for these mitocans to induce apoptosis, they required a functional complex II and III as the pro-apoptotic effects of PST analog SVTH-6 were abolished with the inhibition of ETC complexes II and III. This indicates that SVTH-6 must interact with complex II or III in order to exert its anti-cancer effects. It may be possible that PST analogs interact or bind with complex II or III in order to exploit an unidentified mechanism of cancer cell metabolism or potentially affect downstream pathways of these complexes. Another PST analog we analyzed, SVTH-7, was more efficacious than gemcitabine and any other anti-cancer agent on several cancer cell lines with EC_50_ in the nM range.

Sorafenib (clinical trial number: NCT00105443) has completed phase III clinical trials and MitoTam is expected to begin clinical trials shortly.

### 3.5. Oxidative Stress Targeting

It follows that if mitochondrial OxPhos is targeted, mitochondrial stress may lead to increased ROS production, leading to oxidative stress. Increased oxidative stress can affect mitochondrial proteins, leading to further dysfunction of mitochondria, resulting in generation of additional ROS in a positive feedback loop. Targeting mitochondria or oxidative stress defense mechanisms can initiate this cycle, ultimately leading to cell death [152].

The use of pro-oxidant agents allows for increasing ROS in cancer cells to cytotoxic levels, leading to apoptosis. Cancer cells thrive under the presence of ROS for proliferative signaling, increased mutation rate to further tumourigenesis, DNA damage, and genome instability [41,42,153]. Thus, cancer cells may already have higher, but sublethal oxidative stress presenting a selective vulnerability not found in normal healthy cells. The manganese porphyrin, manganese (III) meso-tetrakis N-ethylpyridinium-2-yl porphyrin (MnTE-2-PyP^5+^), is able to respond to increased ROS levels in cancer cells to modulate the mitochondrial environment and enhance the effects of chemotherapeutics dexamethasone and 2-DG [116]. Rotenone, a complex I inhibitor, activates NOX2 through the PI3K/Akt/mTORC1 signaling pathway, resulting in the release of excessive ROS generation and increased cancer cell death [117]. Similarly, lonidamine induces ROS generation through complex II to promote cell death [118]. Metformin, discussed previously, also exerts its effects on cancer through the induction of oxidative stress [114]. This could presumably be because of inhibition of complex I and interruption of the ETC, resulting in the leakage of electrons to oxygen [73]. Activation of poly (ADP-ribose) polymerase (PARP) will enhance the PARP-1–ATF4–MKP-1–JNK/p38-MAPK retrograde pathway, leading to the generation of ROS followed by cell death [119]. We have also demonstrated that a novel curcumin analogue, Compound A, showed high efficacy and induced selective apoptosis through the generation of ROS in a variety of cancer cell lines alone and in combination with another pro-oxidant, piperlongumine [120].

In contrast, the inhibition of proteins implicated in cancer cells oxidative stress defense mechanisms could be targeted. In doing so, cancer cells may lose their ability to thrive under environments containing a higher concentration of ROS compared with normal healthy cells. One example is the silencing of nucleus-accumbens-1 (NAC1), which has been shown to facilitate oxidative stress resistance, sensitizing cancer cells to oxidative stress generating chemotherapeutics [121]. Pyrroline-5-carboxylate reductase 1 and 2 (PYCR1, PYCR2) downregulation or silencing could potentially sensitize cancer cells to ROS by inhibiting stress-response protein ribonucleotide reductase small subunit B (RRM2B), a protein implicated in protection from oxidative stress [122]. However, this method of oxidative stress targeting may not be ideal because of the necessity and high expression of ROS-protecting mechanisms in healthy noncancerous cells. Piperlongumine has also been implicated in targeting of glutathione S—transferase PI (GSTP1), which inhibits its oxidative stress protection mechanism, leading to cancer cell death [49,154]. Further research should look at the assessment of resulting toxicity or developing targeted downregulation of cancer cells.

### 3.6. Combination Therapies to Target Multiple Vulnerabilities

By using agents capable of targeting different vulnerabilities of mitochondria, effective treatment approaches using well-tolerated concentrations of chemotherapies may be developed. Targeting multiple facets of cancer cell vulnerabilities will additionally prevent the relapse of cancers thanks to targeting cancer cells in multiple environments. For example, targeting both aerobic glycolysis and OxPhos may lead cancer cells to apoptosis.

Our studies on PST analog SVTH-7 indicated that combination treatment with oxidative stress inducing piperlongumine (PL) led to a more efficacious treatment that dissipated the MMP, decreased oxygen consumption, and induced the release of many apoptogenic factors from the mitochondria [155]. Because of the hyperpolarization of the mitochondria in cancer cells, the PST analogs were able to selectively target cancer, possibly through the processing and uptake of PST analogs in positively charged molecules to be brought to only cancer cell mitochondria. In this case, combining a complex II and III targeting agent to induce oxidative stress synergized greatly with ROS, generating PL as an extremely efficacious and selective treatment for cancer.

In this review, we have additionally highlighted many studies wherein one mitochondrial-targeting agent sensitized cancer cells to another treatment [67,68,69,70,103,104,112,136,137,138,145,146,149,150,151]. The ability of mitocans to synergize well with other therapeutics is crucial because of the many conventional treatment approaches utilizing multi-drug treatments to target multiple vulnerabilities of cancer cells.

## 4. Natural Health Products Targeting the Mitochondria

Natural health products (NHPs) are materials isolated from various food and plant sources that have exhibited medicinal properties [156]. NHPs are complex mixtures of various bioactive components, including mitocans, that have the potential for selective and non-toxic treatment of cancer.

Natural health products can target the same mitochondrial vulnerabilities as described previously with other therapies. Chrysin, a natural flavone from plant extracts traditionally used in herbal Chinese medicine, drastically inhibited HKII binding to VDAC1, resulting in extensive apoptosis in hepatocellular carcinoma [123]. Similarly, deguelin [124] and halofuginone (clinical trial number: NCT00027677) [125], natural compounds isolated from *Mundulea sericea* and *Dichroa febrifuga*, respectively, downregulated HKII leading to apoptosis in in vitro and in vivo studies on lung and colon cancer, respectively. Grape seed extract (GSE) was shown to target ETC complex III, deplete levels of glutathione antioxidant, and result in the loss of MMP in cancer [126]. An interesting review from Sehrawat et al. highlights the usage of various NHPs in chemoprevention to cause mitochondrial dysfunction [157]

We conducted research into developing NHPs as selective anticancer therapeutics with high efficacy. These extracts contain multiple active phytochemicals that could target different biochemical pathways to induce apoptosis in cancer cells. Natural extracts of dandelion root [127], long pepper [128], hibiscus [129], lemongrass, and white tea [130] were shown to be highly effective in inducing apoptosis via the generation of excessive ROS and dissipation of MMP. These extracts were non-toxic to healthy cells and were well-tolerated in animals over a long period of time. Oral administration of these extracts in nude mice xenografted with human tumours resulted in tumour burden reduction. Indeed, the dandelion root extract has moved to clinical trials (clinical trial number: OCT1226, DRE) in Canada.

Research into targeting mitochondria via NHPs highlights the potential of these compounds to be used as safe treatment options capable of also being selective for cancer. Many NHPs may also serve as effective partners in combination treatment to allow for the targeting of many cancer vulnerabilities.

## 5. Conclusions

Mitochondria are central to apoptosis, to the generation of ATP via OxPhos, and as a source of ROS particularly in the event of mitochondrial dysfunction. This presents a very interesting organelle to target in cancer cells for the induction of cell death. The development of mitochondria-targeting therapies is interesting and promising, warranting further investigation of these therapies. Targeting mitochondria would be far more selective than targeting DNA replication and repair because of differences in cancer cell and healthy cell mitochondria. By using a selective treatment, approaches can avoid adverse side effects such as organ toxicity that arise from cytotoxicity to normal healthy cells. Mitochondrial-targeting drugs are able to target mitochondrial vulnerabilities in cancer cells, but not healthy cells. Indeed, several current research studies that have specific cancer vulnerabilities are able to be targeted by small molecules, resulting in mitochondrial depolarization and cell death. The usage of mitocans as an adjuvant with other therapies can serve as a deterrent to cancer drug resistance, leading to the efficient elimination of cancer cells. Oxidative stress vulnerabilities combined with mitochondrial differences in cancer cells present an excellent opportunity to kill cancer cells, using mitocans in combination with pro-oxidative modulators. 

This strategy is in the early stage of development as an anticancer therapy approach, but has the potential to bring transformational change in cancer treatment. Although scientific research indicates that these compounds are generally nontoxic and well-tolerated, more work is needed to observe the long-term effects of mitocans and pro-oxidant usage on the health and physiology of vital organs. Accelerated studies pertaining to safety, efficacy, and human clinical trials are urgently needed to bring some of these compounds to general cancer therapy regimens. 

## Figures and Tables

**Figure 1 cancers-11-00916-f001:**
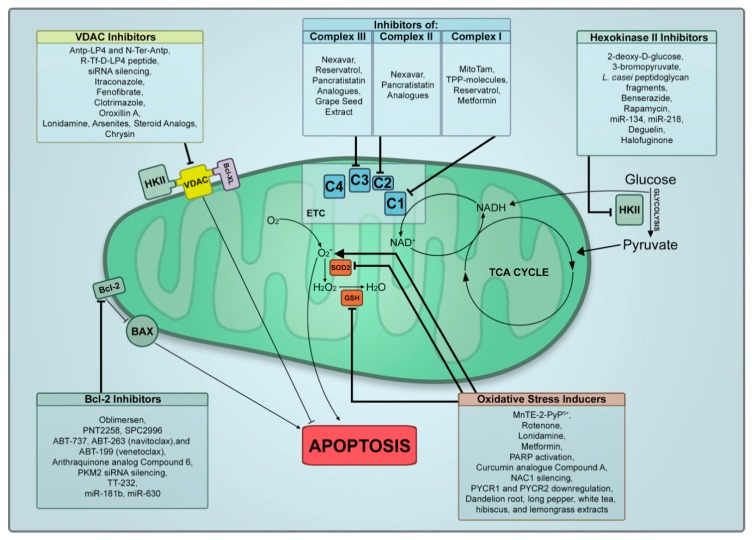
Mitocan treatments and mitochondria targets. Summary of reviewed mitocan treatments and their targets in the mitochondria. HKII, hexokinase II; C1, complex I; C2, complex II; C3, complex III; C4, complex IV; ETC, electron transport chain; VDAC, voltage-dependent anion channel 1; TPP, triphenylphosphonium; PARP, poly (ADP-ribose) polymerase; NAC1, nucleus-accumbens-1; PYCR, pyrroline-5-carboxylate reductase.

**Table 1 cancers-11-00916-t001:** Mitocan treatments, targets, and treatment effects. VDAC1—voltage-dependent anion channel 1; 2-DG—2-deoxy-D-glucose; 3-BPA—3-bromopyruvate; Benz—benserazide; STAT3—signal transducer and activator of transcription 3; HK—hexokinases; PBMCs—peripheral blood mononuclear cells; MMP—mitochondrial membrane potential; PKM2—pyruvate kinase M2 isoform; TPP—triphenylphosphonium; ROS—reactive oxygen species; PARP—poly (ADP-ribose) polymerase; NAC1—nucleus-accumbens-1; PYCR—pyrroline-5-carboxylate reductase; GSE—grape seed extract.

Mitochondrial Targets	Mitocan/Treatment Name	Treatment Effect	Clinical Trial Status	Reference
Hexokinase II	2-DG	Cytotoxicity, sensitization to prednisone	NT *	[73,74]
	3-BPA	Cytotoxicity, sensitization to prednisone	NT	[74,75,76]
	*Lactobacillus casei* peptidoglycan fragments (European Patent number 1217005)	Inhibition of entire metabolism of cancer tumour cells	NT	[77]
	Benz	Reduces glucose uptake, lactate production, and ATP levels, led to apoptosis	Phase 4 (NCT02741947))	[78]
	Rapamycin/siRNA downregulation of STAT3	Glycolysis inhibition, reduce glucose consumption	NT	[79]
	miR-134	Knockdown of HKII reduced glucose consumption leading to apoptosis	NT	[80]
	miR-218	Downregulation of HKII and apoptosis	NT	[81,82]
VDAC-1	VDAC1-based peptides Antp-LP4 and N-Ter-Antp	Highly effective in inducing cell death in leukemia patient PBMCs and cancer cell lines, but not healthy patient PBMCs	NT	[83,84,85]
	R-Tf-D-LP4 peptide	Targeted transferrin receptor in cancer cells, enhancing specificity of Antp-LP4 and N-Ter-Antp	NT	[86]
	VDAC-1 siRNA silencing	Decreased MMP and ATP levels, reducing tumour burden	NT	[87]
	Itraconazole	Inhibition of cell proliferation	NT	[88]
	Fenofibrate	Reprogramming of metabolism and apoptosis in oral carcinomas	NT	[89]
	Clotrimazole	Cytotoxicity, inhibition of glycolysis	NT	[90]
	Oroxillin A	Cytotoxicity, apoptosis, cell cycle arrest, and metastasis inhibition	NT	[91]
	Lonidamine	Cytotoxicity	NT	[91]
	Arsenites	Cytotoxicity	NT	[91]
	Steroid Analogs	Cytotoxicity	NT	[91]
Bcl-2 Family	Oblimersen	Downregulation of Bcl-2, synergy with other treatments	(G3139)	[92]
	PNT2258	Cell cycle arrest, apoptosis in non-Hodgkin’s lymphoma	Phase 2 (NCT02226965)	[93]
	SPC2996	Leukemic cell clearance, immune system activation and stimulation	Phase 2 (NCT00285103)	[94]
	ABT-737	Apoptosis in lymphoma and leukemia cell lines	NT	[95,96]
	ABT-263 (navitoclax) and ABT-199 (venetoclax)	Enhanced effects and specificity compared to ABT-737	Phase 2 (NCT03504644)Phase 2 (NCT03181126)	[97,98]
	Anthraquinone analog Compound 6	Binds Bcl-2, Mcl-2, and p-Mcl-2 leading to apoptosis induction	NT	[99]
	PKM2 siRNA silencing	Regulates oxidative stress induced apoptosis in a variety of cancers	NT	[100,101]
	TT-232	Translocation of PKM2 to nucleus to trigger apoptosis	Phase 2 (NCT00422786)	[102]
	miR-181b	Sensitize cancer cells to cisplatin	NT	[103]
	miR-630	Sensitize cancer cells to cisplatin	NT	[104]
Electron Transport Chain	Sorafenib (nexavar)	Inhibition of ATP synthase leading to Parkin-mediated apoptosis	Phase 3 (NCT00105443)	[105]
	MitoTam	Increased localization of tamoxifen to mitochondria, leading to increased specificity	Clinical trials to begin shortly	[106]
	TPP-Peptide Artemisinin-TPP Green titania ((G-TiO_2-x_) conjugated to TPP	Selectively kill anticancer cells	NT	[107,108,109]
	Reservatrol	Act as a pro-oxidant leading to cancer cell death	NT	[110,111]
	Metformin	Selective mitochondrial targeting, acts as an adjuvant with many cancer therapies	Phase 1 (NCT03477162)	[112,113,114]
	Pancratistatin analogues SVTH-6 and SVTH-7	Highly selective cytotoxicity on cancer cells in 2D and 3D culture models	NT	[115]
Oxidative Stress	MnTE-2-PyP^5+^	Enhance chemotherapeutic effect by mitochondrial environment modulation	NT	[116]
	Rotenone	Activates NOX2 resulting in increased ROS and cell death	NT	[117]
	Lonidamine	Cytotoxicity through ROS generation	NT	[118]
	Metformin	Additionally exert oxidative stress	Phase 1 (NCT03477162)	[112,113,114]
	PARP activation	Enhances ROS production leading to apoptosis	NT	[119]
	Curcumin analogue Compound A	Selective apoptosis through the generation of significant ROS in a variety of cancers	NT	[120]
	NAC1 silencing	Removal of oxidative stress defense mechanism, sensitization	NT	[121]
	PYCR1 and PYCR2 downregulation	Sensitizes cancer cells to ROS by inhibiting stress-response proteins	NT	[122]
Natural Health Products Targeting Mitochondria	Chrysin	Inhibits HKII binding to VDAC1 leading to apoptosis	NT	[123]
	Deguelin	Downregulates HKII leading to apoptosis	NT	[124]
	Halofuginone	Downregulates HKII	Phase 1 (NCT00027677)	[125]
	GSE	Targets complex III and depletes glutathione antioxidant leading to apoptosis in cancer	NT	[126]
	Dandelion root, long pepper, white tea, hibiscus, and lemongrass extracts	Highly effective induction of apoptosis and excessive ROS generation	Phase 1 (OCT1226, DRE)	[127,128,129,130]

* NT = not tested in clinical trials.

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
