# Peer review of "Exploiting Mitochondrial Vulnerabilities to Trigger Apoptosis Selectively in Cancer Cells"

_cancers, 2019, doi:10.3390/cancers11070916_

Round 1
Reviewer 1 Report
In the review “Exploiting Mitochondrial Vulnerabilities to Trigger Apoptosis Selectively in Cancer Cells” Nguyen and Pandey explore the expanding group of compounds and several experimental strategies targeting mitochondrial metabolism and/or biomolecules (protein and mtDNA) to employ them in cancer treatment. This topic is important and a comprehensive presentation of the recent advances in this field will be very useful for readers in many areas of cancer biology.
I recommend this manuscript to be published with minor revisions.
Following please find my comments and suggestions.
1. I strongly suggest that authors should consider supplementing the text with table(s) summarizing targets of mitocans, individual compounds, literature, etc.
2. Similarly, one or two figures would help to communicate in a complex way what is the main focus of this work. I would direct authors to previous reviews in the field (e.g. Neuzil et al., 2013). Authors aim to “…highlight several mitochondrial targets and expand the breadth to include all cancer types.” Would it be possible to depict changing metabolic strategies in various cancer types?
3. Authors should consider hierarchy of sections and subsections. For example, it is unclear why chapter 4.2. Targeting Bcl2 is a subsection of a chapter about targeting HK2, etc.
4. Lines 105-109: Authors should list literature references for their hypothesis stated here.
5. It is important to mention low glucose microenvironment of the ovarian cancer (lines 72 to 77) as the key driver of the metabolic reprogramming.
6. Lines 135-136: “Cancer cells are believed to require a high concentration of ROS to supplement their 135 increased proliferation rates…” Authors shouldn’t miss the opportunity to express their critical opinion on this believe. What is the cause and what is the consequence here?
Minor stylistic points:
7. Throughout the manuscript there are many lines where the language is convoluted and over-inflated. One example for all: Lines 379-380: “ By using agents capable of targeting different vulnerabilities of mitochondria, efficacious treatments able to be used at well-tolerated concentrations may be developed.”
8. Frequent use of phrases like “[the] vicious cycle” should be avoided. Similarly, repetitive writing like line 84: “Cancer cells…” followed by line 85: “Cancer cells…” is not increasing stylistic quality of the manuscript. Another example: Line 140: “The production of ROS in the mitochondria is…” and line 141: “ The production of ROS in mitochondria is…”
9. Importantly, there are numerous typos and obviously missing (or excessive) words in the manuscript. Couple examples: Line 127: “…will result into activation of apoptotic regardless.”; line 143: “…defence system include superoxide dismutase and glutathione peroxidase…”; line 181: “…was overcome by utilizing doxirubicin-loaded micelles…”; line 209: “…3-BPA [effects?] may be dependent on tumour microenvironment and glucose availability…”; line 378: “4.5. Combinations Therapies to Target Multiple Vulnerabilities”; line 365: “In contrast, the inhibition of proteins implicated in cancer cells oxidative stress defence mechanisms could be targeted.”
10. Lines 148-149: “It would be worthwhile combining mitochondrial targeting agents with oxidative stress inducers to target cancer cells.” reads difficultly.
11. Writing and abbreviating Hexokinase 2 (HK2) should be unified.
Author Response
Subject: Submission of revised manuscript cancers-513964
Dear Ms. Sara Radunovic and editors of Cancers,
Thank you for your letter with reviews on our review submitted to Cancers. We would like to thank the reviewers for their time and thorough review of the manuscript. We appreciate the very constructive and important comments/suggestions provided by the reviewers. We have revised the manuscript to address the comments by the reviewers below and we are submitting the revised manuscript. Our responses to each comment can be found further down in this response.
We hope the revised manuscript is now acceptable for publication and we are looking forward to hearing from you soon.
Sincerely,
Siyaram Pandey
Response to Reviewer 1 Comments
1. I strongly suggest that authors should consider supplementing the text with table(s) summarizing targets of mitocans, individual compounds, literature, etc.
Response: Thank you for this valued suggestion, we have added “Table 1” summarizing all mitocans and treatments discussed in the paper and have referenced to it in the section preceding the treatment reviews.
2. Similarly, one or two figures would help to communicate in a complex way what is the main focus of this work. I would direct authors to previous reviews in the field (e.g. Neuzil et al., 2013). Authors aim to “…highlight several mitochondrial targets and expand the breadth to include all cancer types.” Would it be possible to depict changing metabolic strategies in various cancer types?
Response: Thank you for this suggestion. Due to time constraints and not much experience generating figures, we were unable to create our own figure with all mechanisms of mitochondrial vulnerabilities. If you feel that a figure is necessary for the submission of this manuscript, we would be more than happy to see if another student in our group is more adept to design the figure and we would be able to send one.
3. Authors should consider hierarchy of sections and subsections. For example, it is unclear why chapter 4.2. Targeting Bcl2 is a subsection of a chapter about targeting HK2, etc.
Response: Thank you for catching this. In our main manuscript file, our titles were not numbered and so we believe that the number issue is something that would happen when generating the manuscript sent for peer review. We have fixed this issue to make the manuscript hierarchy more sensible.
4. Lines 105-109: Authors should list literature references for their hypothesis stated here.
Response: Thank you for this comment. We wanted to offer a potential reason for why differences in healthy and cancerous mitochondria may exist. There are mitochondrial related changes that can be referred but we have not conclusively proved that this is the case yet.
5. It is important to mention low glucose microenvironment of the ovarian cancer (lines 72 to 77) as the key driver of the metabolic reprogramming.
Response: Thank you for this comment. We have made this addition to the manuscript.
6. Lines 135-136: “Cancer cells are believed to require a high concentration of ROS to supplement their 135 increased proliferation rates…” Authors shouldn’t miss the opportunity to express their critical opinion on this believe. What is the cause and what is the consequence here?
Response: Thank you for this comment. We have made this addition to the manuscript.
7. Throughout the manuscript there are many lines where the language is convoluted and over-inflated. One example for all: Lines 379-380: “ By using agents capable of targeting different vulnerabilities of mitochondria, efficacious treatments able to be used at well-tolerated concentrations may be developed.”
Response: Thank you for your suggestion. We have reviewed the manuscript to simplify convoluted writing.
8. Frequent use of phrases like “[the] vicious cycle” should be avoided. Similarly, repetitive writing like line 84: “Cancer cells…” followed by line 85: “Cancer cells…” is not increasing stylistic quality of the manuscript. Another example: Line 140: “The production of ROS in the mitochondria is…” and line 141: “ The production of ROS in mitochondria is…”
Response: Thank you for catching this. We have gone through and removed excessive repetition of phrasing and repetitive sentence writing.
9. Importantly, there are numerous typos and obviously missing (or excessive) words in the manuscript. Couple examples: Line 127: “…will result into activation of apoptotic regardless.”; line 143: “…defence system include superoxide dismutase and glutathione peroxidase…”; line 181: “…was overcome by utilizing doxirubicin-loaded micelles…”; line 209: “…3-BPA [effects?] may be dependent on tumour microenvironment and glucose availability…”; line 378: “4.5. Combinations Therapies to Target Multiple Vulnerabilities”; line 365: “In contrast, the inhibition of proteins implicated in cancer cells oxidative stress defence mechanisms could be targeted.”
Response: Thank you for your thorough suggestion. We agree with your statements and have reviewed the manuscript to fix all typos and excessive wording.
10. Lines 148-149: “It would be worthwhile combining mitochondrial targeting agents with oxidative stress inducers to target cancer cells.” reads difficultly.
Response: Thank you for your suggestion. We agree that this sentence is a difficult read and have removed it as this is something that we discuss later in the review.
11. Writing and abbreviating Hexokinase 2 (HK2) should be unified.
Response: Thank you for catching this, we have made all mentions of hexokinase II streamlined to either be hexokinase II or HKII.
Reviewer 2 Report
The manuscript entitled “Exploiting Mitochondrial Vulnerabilities to Trigger Apoptosis Selectively in Cancer Cells” by Nguyen and Pandey focuses on strategies to sensitize cancer cells specifically aiming mitochondria as a vulnerable target. The language is clear and the manuscript is straightforward. However, prior to publication, the paper should be further strengthened to increase the enthusiasm and the quality of work. Please consider addressing the major and minor concerns below.
Major:
1- The introduction explaining the mitochondria in cancers, the first section, is very general and lacks details. The cellular mechanisms, such as apoptosis, are regulated by various signaling cascades that function in a cell-type-dependent manner. Thus, the authors should consider explaining basic mechanisms by giving more specific examples as they did in some part of this section (line 72-77).
2- The statement by the authors (Line 102) “In non-cancerous cells, the mitochondria are assembled properly, and are generally difficult to permeabilize” needs a good literature citation as this review relies on the different nature of mitochondria in healthy and cancerous cells. Is this a well-known fact for every cancer type? Every tissue? Please clarify.
3- Section 2.1., please clarify the following statement: “Cancer cells are believed to require a high concentration of ROS to supplement their increased proliferation rates…”. What does “high concentration of ROS” mean? Compared to what? Please specify the intracellular ROS concentration range in cancer cells (cell-type) and normal cells (cell-type).
4- Section 2.2., This section is poorly written and unable to give any scientific details. Please elaborate on how mitochondrial reprogramming is important and what strategies are in use. Please give specific examples.
5- The subject is interesting; however, the text is difficult to follow for the reader.
a. The authors should consider adding a table that summarizes inhibitors/drugs/compounds; their direct targets; how these compounds function; why it works differently in cancer and non-cancer cells. These details together in a table would give a clear picture of physiological relevance.
b. The authors should consider including a figure that clearly indicates different levels of targeting strategies for mitochondria. (e.g. Bcl2 Family, VDACs, TCA cycle, ETC complexes, etc. )
6- What is SVTH-7? Please explain the mechanism of action and indicate its direct target.
7- Section 6. “Conclusions” is poorly written. Please elaborate in detail the importance of the mitochondria-directed therapy strategies, relevance in the clinic and necessary future research.
Minor:
1- Please indicate proteins/genes uniformly (e.g. hexokinase II or Hexokinase 2?)
2- Please correct typos (e.g. line 181). Proof-reading is necessary to improve the flow since the main text has several repetitions.
Author Response
Subject: Submission of revised manuscript cancers-513964
Dear Ms. Sara Radunovic and editors of Cancers,
Thank you for your letter with reviews on our review submitted to Cancers. We would like to thank the reviewers for their time and thorough review of the manuscript. We appreciate the very constructive and important comments/suggestions provided by the reviewers. We have revised the manuscript to address the comments by the reviewers below and we are submitting the revised manuscript. Our responses to each comment can be found further down in this response.
We hope the revised manuscript is now acceptable for publication and we are looking forward to hearing from you soon.
Sincerely,
Siyaram Pandey
Response to Reviewer 2 Comments
1. The introduction explaining the mitochondria in cancers, the first section, is very general and lacks details. The cellular mechanisms, such as apoptosis, are regulated by various signaling cascades that function in a cell-type-dependent manner. Thus, the authors should consider explaining basic mechanisms by giving more specific examples as they did in some part of this section (line 72-77).
Response: Thank you for this comment. We have added some more background information on the basic mechanisms of apoptosis.
2. The statement by the authors (Line 102) “In non-cancerous cells, the mitochondria are assembled properly, and are generally difficult to permeabilize” needs a good literature citation as this review relies on the different nature of mitochondria in healthy and cancerous cells. Is this a well-known fact for every cancer type? Every tissue? Please clarify.
Response: Thank you for this comment. We wanted to offer a potential reason for why differences in healthy and cancerous mitochondria may exist. There are mitochondrial related changes that can be referred but we have not conclusively proved that this is the case yet.
3. Section 2.1., please clarify the following statement: “Cancer cells are believed to require a high concentration of ROS to supplement their increased proliferation rates…”. What does “high concentration of ROS” mean? Compared to what? Please specify the intracellular ROS concentration range in cancer cells (cell-type) and normal cells (cell-type).
Response: Thank you for this comment. We have fixed the statement to indicate that the ROS levels are higher relative to normal cells. Although there is a lot of literature indicating that there is an increase, the specific amount of increase depends on the cancer type and thus we have generalized the statement.
4. Section 2.2., This section is poorly written and unable to give any scientific details. Please elaborate on how mitochondrial reprogramming is important and what strategies are in use. Please give specific examples.
Response: Thank you for these suggestions. We have reworded some aspects of the paragraph to clarify the significance of metabolic changes. As we discuss specific examples later in the paper we have added a sentence to refer readers towards those sections.
5. The subject is interesting; however, the text is difficult to follow for the reader.
a. The authors should consider adding a table that summarizes inhibitors/drugs/compounds; their direct targets; how these compounds function; why it works differently in cancer and non-cancer cells. These details together in a table would give a clear picture of physiological relevance.
Response: Thank you for this valued suggestion, we have added “Table 1” summarizing all mitocans and treatments discussed in the paper and have referenced to it in the section preceding the treatment reviews.
b. The authors should consider including a figure that clearly indicates different levels of targeting strategies for mitochondria. (e.g. Bcl2 Family, VDACs, TCA cycle, ETC complexes, etc. )
Response: Thank you for this suggestion. Due to time constraints and not much experience generating figures, we were unable to create our own figure with all mechanisms of mitochondrial vulnerabilities. If you feel that a figure is necessary for the submission of this manuscript, we would be more than happy to see if another student in our group is more adept to design the figure and we would be able to send one.
6. What is SVTH-7? Please explain the mechanism of action and indicate its direct target.
Response: SVTH-7 is another pancrastitatin (PST) analogue that we analyzed in the same study. Its mechanisms of action and targets are the same as the other PST analog discussed (SVTH-6). We have clarified this in the manuscript.
7. Section 6. “Conclusions” is poorly written. Please elaborate in detail the importance of the mitochondria-directed therapy strategies, relevance in the clinic and necessary future research.
Response: Thank you for this suggestion. We have gone through and made the conclusion more specific at the importance of mitochondria-directed therapies and made a comment on clinic and future prospects of this work. We have discussed the necessity of future research and relevance in the clinic.
8. Please indicate proteins/genes uniformly (e.g. hexokinase II or Hexokinase 2?)
Response: Thank you for catching this, we have made all mentions of hexokinase II streamlined to either be hexokinase II or HKII.
9. Please correct typos (e.g. line 181). Proof-reading is necessary to improve the flow since the main text has several repetitions.
Response: Thank you for your thorough suggestion. We agree with your statements and have reviewed the manuscript to fix all typos and excessive wording.
Reviewer 3 Report
The authors review the recent advances in the development of new therapeutic strategies aimed to exploit the differential mitochondrial function in cancerous respect to normal tissues, given the central role of this organelle in metabolism, apoptosis and reactive oxygen species generation.
In this manuscript, the authors describe the current knowledge on mitocans, small molecules that target mitochondria and modulate the different functions of mitochondria on metabolism, ROS generation or cell death, to selectively dampen cancer cell survival.
This emerging topic is relevant to finding new strategies for cancer treatment and it may be of interest to the readers of the journal.
Overall, the manuscript is well organized and clear, although this reviewer have some suggestions that would help improve the readability:
- The inclusion of a summary table with the mitocans, with their effects on cancer cells would be informative, as is the main interest of this review.
- A figure depicting the different vulnerabilities of cancer cells from the mitochondrial point of view and how can be targeted by the different mitocans also would help the comprehension.
- A list of abbreviations would be helpful.
- The section 2. Targeting mitochondrial vulnerabilities might be better explained, focusing the vulnerabilies of cancer cells to the mitochondrial functions as ROS generation, energy production and apoptosis induction. Then these concepts are used in the targeting strategies.
There are some corrections that should be made:
-The point 4 in line 192 is actually the point 3.1, and therefore all the numbers in the titles after this one should be corrected.
- in line 181 correct "doxirubicin" for doxorubicin
- In the text there are many uses of the comma and "and". In some cases there is no need to put a comma if there is "and" afterwards. Please, revise.
Author Response
Subject: Submission of revised manuscript cancers-513964
Dear Ms. Sara Radunovic and editors of Cancers,
Thank you for your letter with reviews on our review submitted to Cancers. We would like to thank the reviewers for their time and thorough review of the manuscript. We appreciate the very constructive and important comments/suggestions provided by the reviewers. We have revised the manuscript to address the comments by the reviewers below and we are submitting the revised manuscript. Our responses to each comment can be found further down in this response.
We hope the revised manuscript is now acceptable for publication and we are looking forward to hearing from you soon.
Sincerely,
Siyaram Pandey
Response to Reviewer 3 Comments
1. The inclusion of a summary table with the mitocans, with their effects on cancer cells would be informative, as is the main interest of this review.
Response: Thank you for this valued suggestion, we have added “Table 1” summarizing all mitocans and treatments discussed in the paper and have referenced to it in the section preceding the treatment reviews.
2. A figure depicting the different vulnerabilities of cancer cells from the mitochondrial point of view and how can be targeted by the different mitocans also would help the comprehension.
Response: Thank you for this suggestion. Due to time constraints and not much experience generating figures, we were unable to create our own figure with all mechanisms of mitochondrial vulnerabilities. If you feel that a figure is necessary for the submission of this manuscript, we would be more than happy to see if another student in our group is more adept to design the figure and we would be able to send one.
3. A list of abbreviations would be helpful.
Response: We would be happy to provide a list of abbreviations, but Cancers author guidelines do not indicate a location to place these abbreviations. Please let us know if there is any area that would be best to place this table.
4. The section 2. Targeting mitochondrial vulnerabilities might be better explained, focusing the vulnerabilies of cancer cells to the mitochondrial functions as ROS generation, energy production and apoptosis induction. Then these concepts are used in the targeting strategies.
Response: Thank you for this helpful comment. We have reworded this section and have placed a focus on vulnerabilities of cancer cells to mitochondrial functions.
5. The point 4 in line 192 is actually the point 3.1, and therefore all the numbers in the titles after this one should be corrected.
Response: Thank you for catching this. In our main manuscript file, our titles are not numbered and so we believe that this is something that would happen when generating the manuscript sent for peer review. We have fixed this issue.
6. in line 181 correct "doxirubicin" for doxorubicin
Response: Thank you for catching this typo. We have corrected it in the revised manuscript.
7. In the text there are many uses of the comma and "and". In some cases there is no need to put a comma if there is "and" afterwards. Please, revise.
Response: Thank you for your comments. We went through the manuscript and have made all appropriate changes.
Round 2
Reviewer 2 Report
Major
1- The statement (Line 144-146): "In non-cancerous cells, the mitochondria are assembled properly and are generally difficult to permeabilize. However, in cancer cells, mitochondria could be vulnerable to certain agents that can permeabilize the membrane [32]" is not only misleading but also lacks scientific proof. The review article they cited [32] does not characterize or explain a difference about properly assembled mitochondrial membrane or difficulty in permeabilization among cancer and healthy cells. The title of this manuscript is "Exploiting Mitochondrial Vulnerabilities to Trigger Apoptosis Selectively in Cancer Cells" and the authors' hypothesis relies on the mitochondrial membrane permeability/vulnerability in cancer vs healthy cells. Unfortunately, the authors are unable to support this hypothesis with literature. This review might be considered an "opinion letter".
2- As suggested by all the other reviewers, including a Table and a Figure are crucial for this paper because the text has still disconnected parts. I am unable to see any table, contrary to what the authors wrote. Additionally, a basic figure (done in powerpoint or in a very simple program) would visualize the hypothesis and greatly improve the quality of this manuscript. The authors should re-consider adding a simple figure as suggested by all three reviewers.
Minor
I am unsure if the term they often used in the text "Mitochondrial Targeting" sounds right. "Targeting mitochondria" and/or "targeting mitochondrial protein...." would be better.
Author Response
Thank you for all of your valuable suggestions and comments. We have taken them into consideration and revised the manuscript accordingly:
1. The statement (Line 144-146): "In non-cancerous cells, the mitochondria are assembled properly and are generally difficult to permeabilize. However, in cancer cells, mitochondria could be vulnerable to certain agents that can permeabilize the membrane [32]" is not only misleading but also lacks scientific proof. The review article they cited [32] does not characterize or explain a difference about properly assembled mitochondrial membrane or difficulty in permeabilization among cancer and healthy cells. The title of this manuscript is "Exploiting Mitochondrial Vulnerabilities to Trigger Apoptosis Selectively in Cancer Cells" and the authors' hypothesis relies on the mitochondrial membrane permeability/vulnerability in cancer vs healthy cells. Unfortunately, the authors are unable to support this hypothesis with literature. This review might be considered an "opinion letter".
Response: Thank you for your constructive comment. We have decided to remove these statements as they are speculation points that we wanted to highlight and are not central to our review.
2. As suggested by all the other reviewers, including a Table and a Figure are crucial for this paper because the text has still disconnected parts. I am unable to see any table, contrary to what the authors wrote. Additionally, a basic figure (done in powerpoint or in a very simple program) would visualize the hypothesis and greatly improve the quality of this manuscript. The authors should re-consider adding a simple figure as suggested by all three reviewers.
Response: Thank you for this suggestion. We have created a figure to add along with the table that was submitted with our last set of revisions. We have added both the table and figure in the revised submission.
3. I am unsure if the term they often used in the text "Mitochondrial Targeting" sounds right. "Targeting mitochondria" and/or "targeting mitochondrial protein...." would be better.
Response: Thank you for your very valuable suggestion. We have changed the terms to read better. We have also asked several others to review the manuscript and edit for corrections and smooth reading.
Round 3
Reviewer 2 Report
The authors have addressed the reviewer comments. The article is ready for publication.